# Healthcare Stakeholders’ Perspectives on Challenges in the Provision of Quality Primary Healthcare for People with Disabilities in Three Regions of Guatemala: A Qualitative Study

**DOI:** 10.3390/ijerph20196896

**Published:** 2023-10-08

**Authors:** Goli Hashemi, Ana Leticia Santos, Mary Wickenden, Hannah Kuper, Chi-Kwan Shea, Shaffa Hameed

**Affiliations:** 1International Center for Evidence in Disability, London School of Hygiene and Tropical Medicine, London WC1E 7HT, UK; hannah.kuper@lshtm.ac.uk (H.K.); shaffa.hameed@lshtm.ac.uk (S.H.); 2Department of Occupational Therapy, Samuel Merritt University, Oakland, CA 94609, USA; cshea@samuelmerritt.edu; 3Blitz Language, Guatemala City 01057, Guatemala; 4Institute of Development Studies, Brighton BN1 9RE, UK; m.wickenden@ids.ac.uk

**Keywords:** primary healthcare, healthcare stakeholder, people with disabilities, Guatemala, low- and middle-income countries

## Abstract

It is estimated that 3.75 billion people worldwide lack access to healthcare services. Marginalized populations, such as people with disabilities, are at greater risk of exclusion. People with disabilities not only face the same barriers as people without disabilities, but also experience a range of additional barriers in accessing healthcare due to a variety of discriminatory and inaccessible environments. These extra barriers exist despite their greater need for general healthcare, as well as specialized healthcare related to their impairment. Few studies have focused on healthcare providers and the challenges they face in caring for this group. This paper describes the perspectives of healthcare stakeholders and reported challenges to the provision of quality primary healthcare for people with disabilities. In-depth interviews with 11 healthcare stakeholders were conducted in three regions of Guatemala. Data were analyzed using thematic analysis. Five sub-themes emerged from the above theme: infrastructure and availability of resources, impairment-based challenges, need for special attention and empathy, opportunity to provide services to people with disabilities, and gaps in training. The results could contribute to the development and implementation of strategies that can improve primary care provision and ultimately access to services for people with disabilities in low- and middle-income countries.

## 1. Introduction

Universal Health Coverage (UHC) is one of the targets of Goal 3, Health, of the Sustainable Development Goals (SDGs) set in 2015 by United Nations member states to be met by 2030 [1]. The achievement of this goal necessitates the provision of quality and accessible care that is equitably distributed and would not financially jeopardize the user’s livelihood due to out-of-pocket costs [1]. Primary healthcare has been recognized as an essential part of achieving UHC and health for all, as it both serves as the first point of contact with the national health system for individuals, families, and communities, and contributes to the prevention and management of chronic conditions [2,3]. Access to quality primary healthcare services therefore contributes to minimizing healthcare costs and increasing human capital by reducing episodes of chronic conditions and the development of more complex conditions that could be prevented, potentially resulting in a greater morbidity and mortality [2,3].

It is estimated that about 3.75 billion people worldwide, at least half of the world population, lack access to healthcare services, including primary healthcare [4]. Marginalized populations, defined by age, gender, income, ethnicity, sexual orientation, and disability, are at a greater risk of exclusion from healthcare than others [2]. Among these, people with disabilities are one of the largest and poorest minority groups in the world, making up 16% of the world population [4]. They not only face the same barriers, such as cost and geographical barriers, in accessing healthcare as people without disabilities but also frequently experience additional barriers and exclusions due to a variety of discriminatory and inaccessible environments [5,6]. These greater barriers exist even though people with disabilities on average have a greater need for general healthcare, as well as specialized healthcare related to their impairment [7,8,9]. Consequently, people with disabilities face poorer access to healthcare. For instance, research from countries at all income levels demonstrates that people with disabilities attend fewer routine health examinations and are less likely to receive preventative care in comparison to those without disabilities [6,8,10]. There is also evidence that, when people with disabilities seek healthcare, they receive poorer quality services and incur greater expenses [9,11,12]. These healthcare barriers contribute to the worse health outcomes experienced by people with disabilities, including a 10–15-year-shorter life expectancy [13].

Over the past couple of decades, there have been many studies that have looked at the barriers faced by people with disabilities and access to general and primary health care worldwide, from both the perspectives of people with disabilities and healthcare stakeholders [5,6]. A meta-synthesis of 41 qualitative research studies on barriers to accessing primary healthcare by people with disabilities in low- and middle-income countries (LMICs) suggests that accessing healthcare is influenced by three types of barriers: cultural beliefs or attitudinal barriers, informational barriers, and practical or logistical barriers [5]. While the themes were identified as separate, the authors recognized the complexity and dynamic interaction between these themes and how they both have an influence on one another but also have an impact on the supply side (i.e., healthcare sector) and demand side (i.e., people with disabilities) of healthcare experience of people with disabilities. This indicates that the experiences along the health-seeking journey can have an influence on and be influenced by various factors along the process.

Given the complexity of the matter, it is important to include multiple perspectives on the factors that impact access to health and the provision of quality primary health care to people with disabilities. Amongst all the studies, however, few appear to focus on healthcare providers’ perceptions of these barriers and facilitators in providing quality primary health care to people with disabilities. When healthcare providers have been included in the studies, their input has continued to focus on the barriers faced by people with disabilities rather than the challenges they may be facing in providing quality primary healthcare services to people with disabilities [5]. Recognizing the challenges faced by the supply side is just as important as those faced by the demand side. In fact, awareness of factors influencing access to healthcare from both the supply and demand sides will provide us with a more comprehensive understanding of all aspects of the healthcare utilization process and the changes that may be needed at the systemic and policy level. This is of particular concern in LMICs where 80% of the world population reside, yet where the evidence gaps are even more extreme [2].

This paper aims to describe the perspectives of healthcare stakeholders’ on the factors that influence a health practitioner’s ability to provide quality primary healthcare for people with disabilities in Guatemala.

### Context

Guatemala has a population of 17 million individuals. It has the largest economy in Central America, but experiences one of the highest rates of poverty and inequality in the region [14]. The country is administratively divided into 22 departments, with a centralized healthcare system and a constitution that states that every citizen has the right to universal healthcare [14]. Government healthcare spending in Guatemala, however, is one of the lowest in Central America, at 6.21% of GDP in 2019, and it has the highest coverage gaps in basic healthcare, with the lowest healthcare to inhabitant ratio in the region [15].

The public healthcare sector is said to cover approximately 70% of the country’s population [16]. However, it is highly underfunded, with funding gaps resulting in a lack of basic medicine and equipment [17]. These system inadequacies are even more marked in rural areas, which often have large indigenous populations [16,17]. Consequently, those who can afford it seek care from the private/for profit and non-profit sectors. The International Monetary Fund recommends that the short-term goal for Guatemala should be to enhance its primary healthcare coverage with emphasis on prevention, particularly in rural areas where access to services is even more difficult [15].

According to a national survey conducted in Guatemala in 2016, it was estimated that Guatemala’s all-age prevalence of disability was 10.2%, with about 31% of all households including at least one person with a disability [18]. Families that include people with disabilities were more likely to be in the lower socio-economic status groups, have larger household sizes, and have a higher dependency ratio with a lower proportion of household members who were working compared to households without a disabled family member [18]. People with disabilities in Guatemala also face extra direct financial stressors (e.g., healthcare costs and transportation costs), indirect (e.g., reduced participation in the labor market of people with disabilities or their caregiver) and opportunity costs (e.g., time taken away from work to attend appointments), which may further deepen their poverty [2].

Guatemala has a disability law enacted in 1996, called the Law for the Integral Attention to People with Disabilities—Decree 135–96 [19]. It states that people with disabilities are entitled to full integration into society with equal opportunities in areas such as health, education, work, and recreation [19]. Guatemala also ratified the UN Convention on the Rights of People with Disability (CRPD) and its Optional Protocol in April 2009 [20]. The country has been in the process of finalizing a new law on disability, which would mean that it is in greater compliance with the UNCRPD; however, the articles are pending final approval [19,20].

## 2. Materials and Methods

A qualitative study was undertaken in three regions of Guatemala during October 2017 to explore factors affecting access to primary healthcare services for people with disabilities. The regions were: Guatemala City in Guatemala (urban); Tamahu, Coban, in Alta Verapaz (semi-urban and rural); and Santiago Atitlan and Panajachel in Solola (semi-urban and rural). The areas were identified based on recommendations and partnership with local Organizations of Persons with Disabilities (OPDs) working with CBM—Guatemala, a disability-focused international non-governmental organization with a long history of working in the area and their established contacts.

The study involved in-depth interviews with people with disabilities and their caregivers (demand side) and healthcare stakeholders (supply side). The data pertaining to the perspectives and experiences with the Guatemalan healthcare system collected from people with disabilities and their caregivers were reported in the literature by Hashemi et al. (2023) [21]. This paper presented data collected from the healthcare stakeholders’ perspectives.

Interview guides were developed by the primary author, based on the preliminary results of a meta-synthesis of qualitative studies focusing on barriers to accessing primary healthcare services for people with disabilities, and in consultation with disability stakeholders in Guatemala and internationally [5].

Participants were identified through purposive and snowball sampling processes to ensure the sample included a range of primary healthcare providers, administrators, and government stakeholders with various years of experience. Interview questions for healthcare stakeholders inquired about their work experiences, including: (1) the population or sector of health they served, (2) the challenges they perceived are faced by people with disabilities in accessing primary healthcare services, (3) the challenges faced by primary healthcare practitioners in providing quality healthcare services to people with disabilities, and (4) recommendations on how access to primary healthcare may be improved for people with disabilities. The interview guides were translated into Spanish by a registered Guatemalan interpreter to ensure cultural appropriateness. This process involved two teleconferences to ensure that the translated guides would match the original and minimize the amount of content lost in translation.

Interviews were conducted by the principal investigator (G.H.) with support of a Spanish interpreter and research assistant (A.L.S.). The primary investigator, who is a woman, is an occupational therapist trained in Canada and a PhD candidate in public health with extensive travel, clinical, and qualitative research experience in several LMICs. A.L.S. was trained on qualitative interview techniques and the importance of confidentiality and direct translation to minimize changes or misinterpretation during the translation process. While most of the interviews were completed in Spanish, some participants also spoke in English with the principal investigator during parts of the interview. All interviews took place at the participants’ places of employment, lasted for 60–80 min, and were audio-recorded.

### 2.1. Ethics Statement

Ethical approval was received from the Ethics Board at the London School of Hygiene and Tropical Medicine in London, England (Reference #: 17,627 /RR/14983), and The Institucional de Ética-INCAP (Instituto de Nutrición de Centro América y Panama) in Guatemala (Reference #: CIE-REV No. 068/2017) An informed consent sheet in Spanish was provided and reviewed item by item with each of the participants to ensure the protection of the rights of all participants prior to the beginning of interviews. All participants who were approached consented and signed the consent forms.

### 2.2. Data Analysis

Data were analyzed using thematic analysis [22]. Interviews were transcribed and then anonymized, after which they were uploaded into NVivo 12 software for both data management and initial coding by the primary author. In addition to the coding performed by G.H., three randomly selected transcripts were shared with C.K.S., who also coded them independently. This process allowed for further validation of the inductive coding conducted by G.H. of those transcripts and for C.K.S. to become somewhat familiar with the content of the interviews as an additional opportunity for G.H. and C.K.S. to discuss the content and the categorization into themes.

## 3. Results

A total of 11 healthcare stakeholders were interviewed, of which 9 were direct patient care providers, 6 were women, and 1 lived with disability (Table 1). At the time of the interviews, all participants had been in their current role for at least 2 years, with five having over 10 years of experience (Table 2).

The initial inductive coding of the 11 healthcare stakeholder interviews line by line resulted in a total of 53 nodes. The merging of some of the nodes and recategorization resulted in six primary themes, each with sub-themes, related to the overall healthcare stakeholders’ beliefs and perceptions:The perception of challenges faced by primary healthcare providers in the provision of quality care to people with disabilities;Beliefs about people with disabilities and differences in level of disability due to types of impairments;The perception of barriers experienced by people with disabilities in accessing primary healthcare;Beliefs on the role of Organizations of Persons with Disabilities (OPDs) in supporting this population;The perception of people with disabilities’ health status, including sexual and reproductive healthcare needs;Recommendations on how to improve access to primary healthcare for people with disabilities.

Given that the aim of this paper is to describe the perspectives of healthcare stakeholders on the factors that influence a health practitioners’ ability to provide quality primary healthcare for people with disabilities, only theme one and its sub-themes are explored and discussed in detail. It is, however, important to acknowledge that the themes are all interconnected and likely influence and inform one another.

Five sub-themes were identified as part of theme one: infrastructure and availability of resources, impairment-based challenges, the need for special attention and empathy, opportunity to provide services to people with disabilities, and gaps in training.

### 3.1. Infrastructure and Avilability of Resources

This theme recognizes that there are common barriers for all, as well as those that are more specific to people with disabilities. It refers to the lack of resources needed to treat people with disabilities related to equipment, time, space, staffing, or expertise that is beyond the limitations to provide quality care for all service users in general. While initially all participants pointed out the limitations in resources that impact the quality of care for all, such as the lack of medicines and staffing, with additional prompting, they were able to identify and discuss the additional and specific barriers focused on caring for people with disabilities. These barriers can be categorized into two groups: those related to patient’s accessibility needs related to their impairments, such as ramps, equipment, and sign language interpreters, and those related to additional needs, such as extra time and space.

All participants recognized the challenges in working with people with physical disabilities, especially wheelchair users, as demonstrated by the following quote by one of the healthcare stakeholders, identifying the lack of physical accessibility and disability appropriate equipment: *“Services are not adjusted for people with physical disabilities…. No handles. Even the beds, they are like tables…”* (Healthcare stakeholder).

In addition to the need for special equipment, some participants also discussed the lack of human resources and services needed to support people with sensory impairments, such as those with hearing or visual impairments, as expressed by another healthcare stakeholder:


*“…I believe I did not say about interpretation. Especially for deaf people. Sometimes they go on their own and some of them they cannot write…. Maybe a dream to have it but someone to interpret communication. Even for deaf blindness. Adults who have acquired the deaf blindness maybe they can’t see it or hear it—how will they get the information? So need interpretation….”*
(Healthcare stakeholder).

General resources, such as time, are relevant to all potential patients, but may have a greater impact on access for people with disabilities due to the additional barriers they experience. For example, stakeholders discussed how the availability of appropriate transportation can often be limited and may delay a person with disability to arrive at the clinics in a timely manner to receive care before the service quotas for the day are reached.


*“In the centro de salud [Local primary health center] they have different stations. If they [referring to patients with disabilities] come late, sometimes the doctors can’t check them. Or they [the doctors] can only serve a certain number of people and when they come outside those numbers the doctors don’t see them.”*
(Physician).

Another physician explained the spatial challenges and the lack of physical assistive devices in accommodating a patient with physical impairments, potentially causing discomfort to this and other patients sharing the same space:


*“In case of the person who could not control his body [referring to movements]. He used a cane and he needed more space not to bother the people standing next to him. Because he would swing and sometimes it is crowded so he could not be free to move, while he had to wait in the waiting area…. I did not have any extra additional resources to help him or treat him differently…. It is mostly a space problem, starting with the waiting area where he has to wait with people around him and then to come in the doorway is narrow and he needs more space to come in. While in line there is no railing for him to hold on too to keep his balance. So this is mostly a space and (infra)structural problem….”*
(Physician).

### 3.2. Impairement-Based Challenges

In addition to limitations in infrastructure and resources, the participants also evaluated the level of difficulty in providing services to people with different types of impairments. For example, while most of the participants agreed that people with physical or mobility impairments had the most difficulty in accessing primary healthcare, they expressed that caring for individuals with sensory, cognitive, and psychiatric impairments were the most challenging for them. This is exemplified by the quotation below:


*“…the ones who come with problems of intellectual impairment will be harder to treat, because some of the other disabilities they can explain, they can speak what is happening to them, they can say where is the location in their body where they feel pain but people with intellectual impairments don’t like to be in public or with strangers…. When you speak to them, they are not going to understand, if you try to get information from this person, it is not going to happen …. because they don’t understand when you are talking…”*
(Physician).

This physician’s perspective may indicate the experience and comfort level practitioners may have when caring for patients with visible versus invisible disabilities, and the skills needed to interact with those patients who may have communication difficulties. The practitioners may be more equipped to recognize the needs and challenges of those with more visible impairments because their training is more focused on physical impairments rather than on patient interaction and communication.

Ultimately, all participants shared that patients with multiple impairments, resulting in various functional difficulties, are particularly difficult to care for, with the most challenging combination being that of comorbidity between intellectual or psychosocial difficulties with physical impairments.

### 3.3. Need for Special Attention and Empathy

This theme also focuses on the healthcare stakeholder’s views on how the needs of people with disabilities are different from those without disabilities, specifically focusing on the type of attention and service they believe may be needed to provide quality service to this group. This special attention, at times described as specialized care, does not mean the need for impairment-related medical attention but rather refers to customized spaces or specialty trained staff who have greater skills and empathy to work with people with disabilities, as illustrated by the following quotations:


*“According to my personal experience what we need are customized services for people with disabilities. Because sometimes they come with the regular population, and we have a lot of people that are requesting our services as well and the person with the disability they have to wait more than an hour here and with their families they get desperate. They get tired and they don’t receive the attention when they need it. And they are not suffering by themselves, their family and relatives are suffering with them, as they have to carry them and wait with them. They are listening to their complaints. In my personal perspective we need customized services to serve them immediately when they come, because their condition requires certain services at times.”*
(Nurse).


*“…It is not that we do not want to work with them. But I feel that they need more special attention with people who are trained to serve them and feel the empathy with them and how to handle their situations.”*
(Physician).


*“…We have to treat them with a little more empathy. We need to give them more time. If it is a person who does not walk we have to lead them with transportation from one place to another…if we can have more staff we can give a little more human niceness to them, to each patient. We can be more specific with each of them.”*
(Nurse).

The quotations show a consensus that, in order to provide quality primary care to people with disabilities, they need special treatment by the staff, described as “empathy” and “niceness”. This appears to indicate that the empathy required by people with disabilities is above and beyond what the clinician and staff can provide, either due to time or skills, suggesting that specially trained practitioners or staff are necessary to serve patients with disabilities.

### 3.4. Opportunity to Provide Services to People with Disablities

This theme relates to the perceived opportunity or the frequency that the primary healthcare workers felt they have in serving people with disabilities. Some practitioners stated having many opportunities to see and work with people with disabilities. However, others claimed that they hardly saw any patients with disabilities, limiting their experience with this population. While infrastructure, such as the location and accessibility of different facilities, may be a major barrier for people with disabilities to access health centers, it is worth considering that this may have also been skewed by the practitioners’ ability to recognize certain patient challenges as disabilities. For example, while one participant was able to provide examples of people with sensory impairments as having a disability, another’s definition may be limited to only those with physical impairments, as demonstrated by the following two quotations:


*“We have a patient with a hearing impairment who cannot talk-he needs an interpreter and he brings that person with him. Usually a relative. I have had a person with blindness too and he could hear so I explained everything to him. And he could understand…. We have had lots of people with disabilities.”*
(Physician).


*“The problem is when the person reaches a place they [the physicians] make a record only if they see something different. If they see someone using a wheelchair they will record the disability. Otherwise they do not recognize that this person has a disablity…. They only provide information on physical and intellectual impairment, not hearing or visual afflictions. That is not a disability for them.”*
(Healthcare stakeholder).

In addition to the limited understanding of disability held by some of the participants, despite their awareness of the barriers resulting from the physical infrastructure in the community, some clinicians wondered if people with disabilities may be attending other centers for primary care services as opposed to the local health centers. This is demonstrated by the following quotation:


*“…Probably, it’s because the people [with disabilities] are now going to other places. For example I have had a chance to have a couple of shifts at the hospitalito [private hospital] and I have seen that a couple of children with disabilities go there with their parents so the pediatrician is in charge of taking them and check on their health.”*
Physician

The above contradicts the participants’ recognition of the ongoing barriers in the community related to the physical infrastructure.

Some even believed that the reason people with disabilities are not attending primary healthcare centers may be because their primary health care needs are met at their specialized healthcare appointments:


*“Because, I think they have good attention where they are- it maybe a little more specialized, but they receive good services …when they are receiving services for their condition… That is what I think because they are receiving their specialized services and they take advantage of that, since they have specialized services- also requesting assistance for any other condition they may have. I have seen some people who come here with a flu or something that is small, but I don’t see them every day. That is why I think they are receiving their specialized services...”*
(Nurse).

In the above quotation, while the nurse mentions that she has seen people with disabilities attending the clinic with the flu, she appears to feel that specialized services, which are often only available in the capital city, are so comprehensive that people with disabilities are not in need of primary healthcare. While this implies that people with disabilities’ primary healthcare needs may be different from others in the population, it also suggests that people with disabilities may have less need for primary healthcare. This is a misconception, as people with disabilities are in fact in greater need of primary healthcare.

Lastly, some participants felt that their limited opportunity to care for people with disabilities may be due to Organizations of Persons with Disabilities (OPDs) failing to take people with disabilities to the health centers. In this case, the clinicians recognize that the need exists, but the clients are either not referred to the service or are taken elsewhere, thus placing the responsibility on a third party for people with disabilities to access primary healthcare services. This sentiment was stated by a physician:


*“Around 3 years ago we received a lot of people with disabilities. I understand it was a coordination between the city hall and ADISA [an OPD] and they were taking care of the people with disability and brough them here…. but up to this time now I have not received people with disabilities. I think it is the fault of the coordination or alliances between organizations who were bringing people here in order to be checked…”*
(Physician).

The above may be true, and there may be partnerships between certain agencies that facilitate access to primary healthcare services for people with disabilities. However, it also implies that people with disabilities themselves may not have agency in the decision making as to whether they seek and where they seek primary healthcare services, which is a fallacy, as discussed in detail in Hashemi et al. (2023) [21].

### 3.5. Gaps in Training

Gaps in the training of healthcare workers about disability (e.g., knowledge, skills, and behaviors) were identified as another challenge to the provision of quality primary healthcare to people with disabilities, as discussed in the following quotations:


*“Staff are not fully trained for people with disabilities …a lot of doctors don’t know how to deal with persons with disabilities. They are deficient with disabilities. For example, I have seen that the doctors talk more to the accompanier and not the patient.”*
(Healthcare stakeholder).


*“I believe that a limitation is human resources and staff are not trained. I don’t know what the answers has been of other people but in this hospital, we provide the same services for people with and people without disabilities. It is the same. One of the reason’s we provide same services is that we, the medical staff have not been trained to provide services to a person who comes with disabilities.”*
(Physician).

What the above quotations imply is that the practitioners lack skills and knowledge on how to interact with people with disabilities and that treating people with disabilities the same as people without disabilities is insufficient.

Finally, in addition to the training for healthcare providers and staff, one participant also mentioned the lack of training and education of family members as a barrier to providing quality care to people with disabilities:


*“The thing is if the person comes a little sicker, a little more serious, acute services need to be provided then we need to implement more urgency seeing this person with disability…. Yes that usually happens due to lack of training and education. The families sometimes don’t know when is the right moment to bring a patient to seek assistance…. And some sometimes there is a situation that they are illiterate and they don’t speak Spanish and that makes it a little difficult so they don’t about the signs and symptoms the patient is having until it is too serious and then they come to us.”*
Nurse

The prevalence of such cases, as stated in the quotation, may be greater for people with disabilities due to the myriad of other barriers they face, such as access to education and the complexity of the decision making to seek or not seek primary healthcare services. In addition, they may also be disincentivized to seek care from the healthcare clinic due to previous poor experiences.

## 4. Discussion

The successful delivery of healthcare services, including primary healthcare, depends on a complex interaction between both the demand and supply sides of the healthcare process. While the barriers faced by people with disabilities in accessing primary healthcare services has been extensively studied, little has been published on the challenges faced by primary healthcare practitioners in providing quality care to this population [5,6]. This study aimed to identify the challenges faced by primary healthcare practitioners in three regions of Guatemala. The results of the study identified five themes: infrastructure and lack of resources, impairment-based challenges, the need for special attention and empathy, opportunity to provide services to people with disabilities, and gaps in training.

As expected, a common challenge identified was the poor physical infrastructure of the health facilities and the lack of resources, both general (e.g., time, space, and staff) and impairment-related (e.g., sign language interpreter and accessible beds). While gaps in resources can affect all patients, it was recognized that they can have a greater impact on the accessibility of services by people with disabilities and can make serving people with disabilities, particularly of certain impairments, significantly more difficult. This finding is in line with the results of studies looking at the barriers faced by people with disabilities from the perspective of both people with disabilities and healthcare providers [5,23,24].

What stood out in this study, however, were two sentiments expressed by healthcare providers: one, the practitioners appeared to feel the necessity for separate spaces and specially trained staff, with a particular focus on how to interact with people with disabilities; and two, the need for “empathy” in order to provide quality primary care services to people with disabilities. The first sentiment is somewhat similar to the findings from Tri Handoyo et al. (2021), who found that key professionals seemed less favorable to the inclusion of people with intellectual disabilities (IDs), particularly with severe IDs, in Indonesia [25]. While this approach may seem practical at first, it may result on segregation. This, combined with the need for specially trained staff and a focus on empathy, resembles the charity model of disability. This model of disability considers people with disabilities as victims of their impairment, who are suffering, in need of special protection and services, and often in separate spaces, resulting in social exclusion and segregation [26]. This model of disability is no longer widely accepted, having been replaced with more rights-based models, such as the International Classification of Functioning, Disability and Health (ICF), which is based on a biopsychosocial model of disability. The ICF model is based on an integration of the social and medical models of disability and recognizes that disability results from the dynamic interaction among a health condition and environmental and personal factors [27].

Furthermore, while the results indicate that the healthcare providers believe that they did not often see patients with disabilities due to them either attending other centers to have their needs met or the lack of partnership between the health center and OPDs, it may also be due to their inability to recognize certain impairments that may be less visible and their limited understanding of disability. In either case, their limited understanding of disability has significant implications for all, as it can impact the identification and need for resources and their allocation. This also results in a further exclusion and lack of appropriate interventions for people who may have less identifiable impairments. It also ignores or minimizes the complexity of factors that contribute to a person with disabilities’ decision-making process involved in seeking primary healthcare services [21].

Overall, the results of this paper suggest that the understanding of disability may not be consistent amongst healthcare providers and that some primary healthcare providers in Guatemala may not have recognized the importance of ensuring people with disabilities have the same access to healthcare as those without disabilities. These perceptions likely stem from a lack of education and cultural beliefs about disability, stemming from the charity model of disability, which unfortunately dominates many communities and cultures. This has significant negative implications for people with disabilities and the community as it reinforces the belief that people with disabilities may be a burden to their families and society due to their dependency [28,29]. Charity and pity can also be oppressive, resulting in the decreased self-esteem and disempowerment of people with disabilities and internalized oppression and loss of human capital by reducing human engagement, productivity, and skill [28,29].

Just as healthcare providers are not immune to the cultural biases and beliefs of the environment they live in, they too can play a role in further shaping cultural biases. This, in addition to the importance to providing quality healthcare and the need for improved knowledge, skills, and better interactions with people with disabilities, makes it imperative that additional training be provided, with a focus on disability identification, disability rights as human rights, and how to interact with this population. Training should be provided not only during medical and nursing training, but also as part of in-service updates [30]. There are a variety of methods that can be used for such trainings considering the context and the type of resources, including engagement with people with disabilities themselves, a model that has proven to be an effective strategy in training medical professionals [30].

In summary, the policy implications of this study include the need for Guatemala to expedite its approval of the new and updated disability law and ensure awareness and education regarding the law and its impact to all stakeholders, including consumers. This will help Guatemala to be more in line with the UNCRPD. This research also reinforces the need for policy makers to consider investing in strategies that reduce barriers to accessing primary healthcare services from both the supply and demand sides.

### Strengths and Limitations

A key limitation of this paper is that this study was conducted in the pre-pandemic period and the healthcare situations in the three regions of Guatemala are likely significantly different. The pandemic resulted in a 5% increase in Guatemala’s poverty rate [31]. This exacerbated the challenges already faced by the healthcare sector. In fact, as of October 2022, only 38% of the Guatemalan population was fully vaccinated [31]. This level of coverage has left large numbers of the population, particularly the more marginalized, such as people with disabilities, more susceptible to infection and hospitalization [31]. It is thus anticipated that the challenges in the provision of quality primary healthcare for people with disabilities may have increased for healthcare practitioners.

In addition to the timing of the study, it would have been beneficial to have had more representation from younger practitioners to compare approaches to disability between them and more experienced clinicians.

The study also has a number of strengths. The focus on the perceptions of healthcare stakeholders about the challenges they face when providing care for people with disabilities is novel. Moreover, we interviewed a variety of stakeholders, including direct healthcare providers.

## 5. Conclusions

The successful delivery of any healthcare service, including primary healthcare, depends on the interaction between both the demand and supply sides of the healthcare process. This also applies when considering access to primary healthcare for people with disabilities. There is a need to focus on both the challenges and access barriers faced by people with disabilities and healthcare providers. The themes identified in this study can complement and contribute to the current literature on barriers to accessing quality healthcare for people with disabilities by focusing on the healthcare providers’ perspectives. While the healthcare services in Guatemala are meant to be inclusive, the charity model appears to dominate the perspectives of healthcare providers in Guatemala, resulting in their dissatisfaction with the system of care. The provision of education on human rights approaches and the inclusion of people with disabilities could be an approach to address and develop strategies to support healthcare providers in their provision of care to people with disabilities.

## Figures and Tables

**Table 1 ijerph-20-06896-t001:** Distribution of the participants based on their gender and roles.

Role	Male	Female	Total
Physician	3	1	4
Nurse	0	3	3
Social worker	0	1	1
Other healthcare stakeholder	2	1	2
**Total**	**5**	**6**	**11**

**Table 2 ijerph-20-06896-t002:** Distribution of the participants based on their gender and years of experience.

Years of Experience	Male	Female	Total
Less than 5 years	2	0	2
5–10 years	2	2	4
Greater than 10 years	1	4	5
**Total**	**5**	**6**	**11**

## Data Availability

The data presented in this study are available upon request from the corresponding author. The data are not publicly available due to the transcripts used as the source of data representing roles that can be easily identifiable based on details of their characteristics.

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
