# Peer review of "Healthcare Stakeholders’ Perspectives on Challenges in the Provision of Quality Primary Healthcare for People with Disabilities in Three Regions of Guatemala: A Qualitative Study"

_ijerph, 2023, doi:10.3390/ijerph20196896_

Round 1

Reviewer 1 Report

This is an interesting article presenting rich data on the experiences of primary health care professionals with people with disabilities in Guatemala. The article presents a review of relevant literature, a rationale for the study, and a detailed design. The findings and discussion need some additional work before considering it for publication. 

Theme 3.2 “impairment specific barriers” could be re-worded as “accessibility barriers” or “disability awareness”, which aligns better with the social model stance taken by the authors. The discussion of this theme (page 9, lines 434-434), could also address the lack of accessibility instead of the “impairment specific needs”

The theme “opportunity to care” could be reworded as “opportunity to provide services”. Care is a controversial concept as it assumes a paternalistic attitude towards people with disabilities who need to be taken care of. While this article is about “healthcare” it would be more consistent with the approach of the authors to state “provide healthcare services” so disabled people are considered on an equal basis to other non-disabled people receiving health care services. See also page 9, line 430, and page 10, line 436. 

Clarify that the ICF is based on, and not the same as, the bio-psycho-social model of disability.

The discussion on the disability definition and impairment (page 10, lines 458-466) needs to be reworded. Clarify the difference between disability and impairment and consistently use the terms. This definition could be foregrounded at the start of the paper to provide clear understanding of the terms. 

The quote on page 8, lines 349-353 does not clearly illustrate the contradiction noted in line 347 or the persistence of barriers in the community. The quote and comment need to be coherent. 

Quote on page 7 (lines 296 onwards) is not very different from the quotes provided under the theme 3.1. infrastructure or provision of special spaces. The following two quotes speak to other aspects such as empathy and human treatment. 

The conclusion of theme 3.3 is a bit unclear (lines 316-320)

Perhaps a better word would be "understanding of disability" (page 8, line 343)

Number 6 missing before the theme title (page 5, line 206)

Identification of quotes is inconsistent, all of them should have the role of the interviewee at the end of the quote.

3.2. typo in the heading "esecific" (page 6, line 264)

Use of italics is inconsistent (page 6, lines 270-277)

The article needs to be proof read as there are several typos throughout. 

Reviewer 2 Report

Thank you for the opportunity to provide feedback. I have a few suggestions to enhance the clarity of the manuscript:

1.      Consider omitting "A qualitative study" from the topic, as it may not be necessary.

2.      Rewrite the opening sentence of the introduction section for better clarity.

3.      On page 2, line 50, please provide an explanation or clarification for the term "same barriers."

4.      Revise the statement about Guatemala's disability law for improved clarity: “Guatemala, has a disability law, the Law for the Integral Attention to People with Disabilities- Decree 135-96, enacted in 1996.”

5.      The manuscript lacks a synthesis of the literature conducted in the context of health care issues in Guatemala. Please add a paragraph summarizing relevant literature.

6.      Correct the typo "esecific" to "specific." It is preferable to use precise terminology rather than vague terms like “specific”.

7.      Again, instead of using "special" in the title, please use the exact phrase that accurately represents the content.

8.      Consider adding a paragraph on the policy implications of the study's findings for healthcare services for people with disabilities. This addition could enhance the manuscript's quality.

 Overall, the paper is well-written, and I found the reviewing this manuscript enjoyable.

Reviewer 3 Report

Thank you for the opportunity to read and review your paper. This is a well written paper that contributes to the body of evidence on the healthcare providers’ perceived barriers and suggested strategies to improve access to primary healthcare for people with disabilities in Guatemala.

Overall comments:

The paper would benefit from an editorial check

Some sentences are very long, please consider dividing (examples: Lines 343-348 or lines 468-472)

While throughout the paper the correct phrasing is used ‘people with disabilities’ there are a few instances of incorrect phrases being used ‘disabled people’

Specific comments:

Introduction, and then again in Discussion

The concept of ‘human capital’ is used without explaining how authors understand it and why it is used in this context?

For example, in Introduction, it is not self-explanatory how access to quality primary healthcare services contributes to increasing human capital.

Or in discussion, the sentence ‘Charity and pity can also be oppressive, resulting in decreased self-esteem and disempowerment or people with disabilities resulting in internalised oppression and then loss of human capital’ is not clear.

Materials and methods

Line 139 This paper presents data collected from healthcare stakeholders.

Data analysis

This section would benefit from a more detailed description of thematic analysis. It is not clear how inductive coding resulted in ‘categorization into themes’. Please describe steps taken during the analysis, including, how categories and themes were formulated and what do the themes reflect.

Results

It is confusing to the reader to see themes generated for the whole study. In this paper, only report themes and sub-themes that are relevant to your paper, where you analysed perceptions of healthcare providers of barriers experienced by people with disabilities, and by healthcare providers, and strategies suggested by healthcare providers.

Also, the first part of the results section where you describe the number of nodes, themes and sub-themes belongs in methods rather than results. But, if using that description, need to tailor it to this particular paper (not the whole study).

3.1 through 3.6 are sub-themes of a theme ‘perception of challenges faced by primary healthcare providers in provision of quality care to people with disabilities’. Correct throughout.

good quality English language

Reviewer 4 Report

Dear author(s), 

This is a very interesting and coherent manuscript. I have to congratulate you for the idea and the research you have conducted. 

My major concern regarding your research is the year of 2017. I know that you mentioned the year as a limitation of your study but I highly reccommend updating the situation of Guatemala after COVID-19 for the disabled persons in no more than a paragraph. IYou may emphasize that Covid-19 put a lot of pressure on the health systems around the world and implicitly in Guatemala. Thus, it is possible for the identified issues to be more frequently met or others may arise. 

In other words, maybe some identified issues remain, and some of them got worse or were partially solved. Five years is a long time and there is a high possibility that changes may take place. 

Thank you and good luck!

Round 2

Reviewer 4 Report

No futher suggestions